# Minimizing MRONJ after Tooth Extraction in Cancer Patients Receiving Bone-Modifying Agents

**DOI:** 10.3390/jcm11071807

**Published:** 2022-03-25

**Authors:** Gal Avishai, Daniel Muchnik, Daya Masri, Ayelet Zlotogorski-Hurvitz, Liat Chaushu

**Affiliations:** 1Department of Oral and Maxillofacial Surgery, Rabin Medical Center—Beilinson Hospital, Petach Tikva 49414, Israel; dannymu@gmail.com (D.M.); dayama@clalit.org.il (D.M.); ayelet.zlotogorski@gmail.com (A.Z.-H.); 2Department of Oral and Maxillofacial Surgery, The Maurice and Gabriela Goldschleger School of Dental Medicine, Tel Aviv University, Tel Aviv 69978, Israel; 3Department of Oral Pathology and Oral Medicine, The Maurice and Gabriela Goldschleger School of Dental Medicine, Tel Aviv University, Tel Aviv 69978, Israel; 4Department of Periodontology and Oral Implantology, The Maurice and Gabriela Goldschleger School of Dental Medicine, Tel Aviv University, Tel Aviv 69978, Israel; liat.chaushu@gmail.com

**Keywords:** MRONJ, dental extraction complications, oral surgery in oncology

## Abstract

Background: Medication-related osteonecrosis of the jaws (MRONJ) is a mucosal lesion of the maxillofacial region with necrotic bone exposure. MRONJ is believed to be multifactorial. Tooth extraction is debatably a risk factor for MRONJ. The targets of the present study were to examine MRONJ occurrence in patients using bone modifying agents (BMAs) for oncology indications and undergoing a dental extraction, and to assess whether suspected predisposing factors can predict MRONJ. Materials and Methods: This retrospective, cohort study included all patients fitting the inclusion criteria and a large tertiary medical center. Data were obtained from the hospital’s medical records using a structured questionnaire. Results: We performed 103 extractions on 93 patients. Local inflammation/infection of the extraction site was most associated with a complication (*p* = 0.001) OR = 13.46, 95% CI = (1.71, 105.41), OR = 13.5. When the indication for extraction was periodontal disease, vertical root fracture, or periapical pathosis, the odds of developing MRONJ were 4.29 times higher than for all other indications (*p* = 0.1), OR = 4.29, 95% CI = (1.16, 15.85). A significant association was found between the time of onset of BMA treatment and time of extraction and the development of MRONJ, OR = 3.34, 95% CI = (1.01, 10.18). Other variables did not correlate with the development of MRONJ. Conclusion: Local inflammation/infection and onset of BMA treatment prior to extraction yield a 10.23 times higher chance of developing MRONJ following tooth extraction. Future protocols should use this information to minimize MRONJ incidence.

## 1. Introduction

Bone metastases are found in patients with advanced cancer; they are present in 70–80% of patients with breast or prostate cancer and 30–40% of patients with lung or other solid tumors [1,2,3]. Bone modifying agents (BMAs) such as bisphosphonates (BPs) and denosumab (Dmab) are widely used to treat skeletal-related events (SREs) such as pathologic fractures, spinal cord compression, neurological deficits, and hypercalcemia. SREs caused by bone metastases from solid tumors negatively affect patients’ quality of life. BMAs are also used for the management of lytic lesions related to multiple myeloma and to treat hypercalcemia associated with malignancy [4,5]. BMAs inhibit osteoclast differentiation and function and increase apoptosis, thereby leading to decreased bone resorption and remodeling. BPs bind the mineral component of bone and interfere with the actions of osteoclasts, while Dmab is a monoclonal antibody that binds and inhibits rank ligand (RANKL), leading to the inhibition of osteoclast formation, function, and survival, which are associated with bone resorption [6,7,8]. Several other medications, such as tyrosine kinase inhibitors, monoclonal antibodies (including tocilizumab), mammalian target of rapamycin inhibitors, radiopharmaceuticals, selective estrogen receptor modulators, and immunosuppressants, have also been implicated in osteonecrosis of the jaws [9]. 

Medication-related osteonecrosis of the jaws (MRONJ) is a mucosal lesion of the maxillofacial region, which presents with necrotic bone exposure, pain, and purulent discharge when infection occurs [10,11,12,13]. Sensory nerve deficit, such as is the case in numb chin syndrome, is also a symptom of MRONJ [14]. 

MRONJ is thought to be multifactorial [15,16]. The differential predisposition of the jaws to MRONJ compared to other bones may be explained by an increased remodeling rate [17,18].

The incidence of MRONJ after tooth extraction in patients with cancer exposed to IV BPs ranges from 1.6% to 14.8% [13]. Tooth extraction has been considered a significant risk factor for MRONJ in patients receiving BMAs [13,19]. Conversely, in a large cohort study, simple tooth extraction was not found to be a strong risk factor for MRONJ [20]. Often, dental extraction cannot be avoided, as an inflamed tooth is also a risk factor [21]. It has been proposed that pre-existing inflammation is the risk factor for MRONJ, rather than the surgical procedure (tooth extraction) itself [22,23], and, therefore, extraction of a problematic tooth may reduce the occurance of MRONJ [13,24]. Some studies demonstrated a low incidence of MRONJ after dental extraction in high-dose BMA patients (0–2.8%) [24], while other reports were as high as 25.2% [25]. 

Another approach to minimizing MRONJ is the cessation of BMA treatment a few months prior to surgical intervention (drug holiday). Clinical results are contradictory [13,26], although an animal model found reductions in the frequency and severity of MRONJ after a drug holiday [27]. The use of platelet concentrates was suggested as a treatment modifier following dental extraction in high-dose BMA patients; this method, although promising, was not shown to reduce the occurrence of MRONJ in high-dose BMA patients undergoing dental extraction [28]. Platelet-rich fibrin (PRF) was shown to have superior results regarding hemostasis in patients receiving oral antiplatelet medications after dental extraction, but it is still controversial in oncology patients [29]. 

The purpose of the present study was to assess the ability of various suspected predisposing factors to predict MRONJ occurrence in patients taking BMAs for oncology indications following tooth extraction. The effectiveness of the drug holiday approach is also assessed. 

## 2. Materials and Methods

### 2.1. Study Cohort

We conducted a non-randomized, retrospective, cohort study, which included all patients fitting the inclusion criteria who visited the Department of Oral and Maxillofacial Surgery at a large tertiary medical center (Rabin Medical Center–Beilinson Hospital, Petach-Tikva, Israel) between 2013 and 2020. Data were obtained from the hospital’s medical records by cross-referencing the specific drugs administered at the oncology department and outpatient visits at the Oral and Maxillofacial Department clinic.

Dental extractions were performed by several clinicians with different levels of experience; no standard protocol for extraction was used. All clinicians in our department followed a similar approach for extraction in BMA patients, including minimal trauma to the jawbone, careful curettage of the extraction socket with removal of periapical pathoses, no alveolectomy, and closure with primary intent. 

The inclusion criteria were: Cancer patients treated with high-dose BMAs such as pamidronate (Aredia), zoledronic acid (Zomera), or denosumab (Xgeva);Underwent a dental extraction at the Oral and Maxillofacial Department clinic between 2013 and 2020;Had a minimum documented follow-up period of 1 year.

The exclusion criteria were: Patients for whom post procedure follow-up was absent or less than 1 year;Patients who received previous orofacial radiation;Patients on chronic BMAs for osteoporosis.

### 2.2. Data Collection 

Data were collected from the patient records using a structured form. 

Demographic data included: Sex;Age;Relevant medical diagnoses which might hinder healing (diabetes and/or anemia);Relevant consumed chronic medications that might hinder healing (anticoagulants, steroids, metformin);Cancer type and BMA treatment type.

Dental extraction data included: Indication for extraction;Number of teeth extracted;Existing local inflammation/infection. Each patient was categorized as having pre-extraction local inflammation/infection if at least one of the following signs existed: clinical: prominent gingival swelling or redness; purulent discharge; presence of a sinus tract; radiographic: periapical or peri radicular radiolucency; vertical alveolar bone loss >4 mm from the CEJ; furcation involvement in multirooted teeth; root fractureTime discrepancy between extraction and beginning of BMA treatment;Drug holiday—discontinuation of BMA treatment before extraction and duration of cessation.

Post-procedure data included: Post-procedure antibiotic treatment and/or oral antiseptic mouthwash;Follow-up period (months);Development of MRONJ (yes/no, primary outcome parameter).

### 2.3. Data Analysis

All analyses were performed using IBM SPSS Statistics for Windows, version 25.0 (IBM Ltd., Armonk, NY, USA).

To examine the relationship between categorical variables and the development of MRONJ post-dental-extraction, chi-square, Tukey–Kramer, and odds ratio probabilities were calculated at a significance level of 0.05.

To assess the effect of the different risk factors in the development of MRONJ, a multivariate logistic regression was structured and odds ratios (ORs) were calculated at 95% confidence intervals. 

## 3. Results

### 3.1. Demographic Data

Ninety-three cancer patients were included. Table 1 presents the demographics and medical backgrounds of the patients included. Twenty-one (29.17%) were men and seventy-two (70.83%) were women. Age ranged between 32 and 84 years, with an average of 62.16 ± 11.5 years. Anticoagulant consumption and diabetes (15.05% and 13.97%, respectively) were the most common factors in the patients’ medical history. Metformin and steroid consumption were found in 11.82% and 8.60%, respectively. The most frequent cancer type was breast cancer (63.44%), followed by multiple myeloma (19.35%). Lung and prostate cancer accounted for 3.23% each, while 10.75% of patients had other cancer types. Other cancer types found include gastrointestinal tumors, neuroendocrine carcinoma, amyloidosis, monoclonal gammopathies, and sarcomas. Of the three BMA types, Zomera was the most commonly administered, in 54 (58.06%) patients, followed by Aredia in 30 (32.26%) and Dmab in 9 (9.68%).

### 3.2. Dental Extraction Data

We performed 103 extractions in 93 patients: 83 (89.25%) had a single extraction and 10 (10.75%) had two. Table 2 presents the dental-extraction-specific data collected. The timing of the dental extraction procedure in relation to onset of BMA administration was calculated and categorized into timeframes: 35 extractions (33.98%) were performed over 6 months after the onset of BMA and 30 (29.13%) were performed 6 months prior to the onset of BMA. Fourteen of the extractions were performed after cessation of BMA treatment for 2 months or more (drug holiday). Forty-three of the extractions (41.75%) were of a single tooth and the rest were of two teeth or more, with twelve extractions (11.65%) of six teeth or more. 

The most frequent indication for extraction was periodontal disease (45, 43.69%), followed by extensive caries (34, 33.01%), periapical pathoses (10, 9.71%), and vertical root fracture (7, 6.8%). 

Sixty-seven extractions (65.04%) presented with a local inflammation/infection. 

Post-extraction treatment included antibiotics only in 2 extractions, antiseptic oral mouthwash in 23 (22.33%), and both antibiotics and mouthwash in 78 (75.73%). 

Follow-up time for all patients was over one year after the extraction procedure, with a maximum of seven years. The average follow-up time was 1.97 ± 1.38 years. 

Twenty (19.42%) of the one hundred and three extraction procedures performed presented a complication of MRONJ in the post-extraction follow-up period. 

In order to examine the correlation between the development of MRONJ after extraction and various variables, the chi-squared test, Cramer’s V test, and odds ratio associations were calculated at a significance level of 0.05. Table 3 and Table 4 summarize the categorical and numerical variables, respectively, in relation to the development of MRONJ. 

Local inflammation/infection of the extraction site was most associated with a complication (*p* = 0.001) OR = 13.46, 95% CI = (1.71, 105.41). Thus, the chance of developing a complication was almost 13.5 times higher among patients with a sign of inflammation or infection compared to patients without. 

Indication for extraction was found to be significantly correlated with MRONJ development. When the indications for extraction were either periodontal disease, vertical root fracture, or periapical pathosis, the odds of developing MRONJ was 4.29 times higher than that of all other indications (*p* = 0.1), OR = 4.29, 95% CI = (1.16, 15.85).

No significant association was found between BMA type and complication development, neither in the division of BMAs into three types of treatment (Denosumab, Aredia and Zomera) nor in the comparison between patients in the Denosumab group and patients in the Aredia and Zomera groups (*p* = 0.3).

A significant association was found between the time of onset of BMA treatment and time of extraction and the development of MRONJ, OR = 3.34, 95% CI = (1.01, 10.18). Patients who took the drug before extraction are 3.34 times more likely to develop a complication. However, starting BMA treatment after extraction does not eliminate the possibility of MRONJ development. 

Variables that did not correlate with the development of MRONJ included sex, age, type of cancer, and history of diabetes or anemia or medications (anticoagulants, metformin, or steroids). Furthermore, factors such as drug holiday, number of teeth extracted, and post-extraction treatment did not show a decreased chance of development of MRONJ.

A multivariate logistic regression using the “enter” method was initially performed on six variables (local inflammation/infection, onset of BMA treatment prior to extraction, periodontal/VRF/PA pathosis indication, previous history of diabetes or anemia, sex, and age) to estimate the chances of developing MRONJ after a dental extraction. The findings of the regression equation showed that the approximate model fit the data (Hosmer and Lemeshow’s goodness of fit = 0.274, sig = 0.029) and that knowledge of the six variables explained over 21% of the variance regarding the probability of MRONJ (Negelkerke R^2^ = 0.214). However, inclusion of all six variables in the regression equation did not leave any of them significant at the 0.05 level. Further running of the equation using the forward LR and backward LR methods suggested that the variables of local inflammation and onset of BMA treatment prior to extraction were the only variables necessary to construct a distinct model, with 10.23 times the chance of developing MRONJ (95% CI: (1.28, 81.69), *p* = 0.028). 

## 4. Discussion

In this retrospective study, we analyzed the factors contributing to the development of MRONJ in patients receiving high doses of BMAs for oncology indications and who underwent a dental extraction procedure. 

Most of the literature to date compares the incidence of MRONJ in BMA patients between those who underwent a dental extraction and those who did not. For instance, Kyrgidis et al. [30], in a case-control study of breast cancer patients exposed to zoledronate, determined that tooth extraction was associated with a 16-fold increase in risk of MRONJ compared to those without ONJ (odds ratio (OR) = 16.4; 95% confidence interval (CI), 3.4–79.6). In a cohort of patients exposed to high doses of bisphosphonates, tooth extraction led to a 33-fold increase in risk of MRONJ [31]. 

The data of the present study showed that tooth extraction may contribute to the development of MRONJ in nearly 20% of procedures. Our findings clearly demonstrate that inflammation/infection is the most influential factor, as it was evident in 95% of cases with MRONJ. However, only 28% of the cases with local inflammation developed MRONJ. When local inflammation/infection was not evident, merely 2.8% of cases developed MRONJ, suggesting that if local inflammation/infection is treated prior to extraction, MRONJ may be avoided in most cases. These results support the findings of Soutome et al. [20]. 

The use of mouthwash alone led to MRONJ in 30% of cases, indicating that using mouthwash postoperatively does not eliminate the possibility of MRONJ. Using both mouthwash and antibiotics postoperatively led to an 18% MRONJ occurrence, suggesting that this has no effect on minimizing its incidence. Therefore, minimizing postoperative inflammation/infection does not reduce the risk of MRONJ. It is possible that preoperative efforts to reduce inflammation/infection may decrease the risk of MRONJ. Future studies should be dedicated to confirming this speculation. 

A drug holiday was not effective in reducing MRONJ, which had an incidence of 50%. As per these results, it cannot be recommended. A similar conclusion regarding the non-effectiveness of drug holidays in cases of high doses of BMAs was recently published in both a systematic review [32] and a randomized clinical feasibility study [33]. These results negate the current concept that drug holidays are effective in reducing the occurrence and severity of MRONJ when extraction is performed after a drug holiday in an animal model [27]. 

BMA treatment prior to extraction led to an MRONJ incidence of 28%. BMA treatment post-extraction led to an MRONJ incidence of 10.4%. It is clear that eliminating inflammation/infection is more important than the timing of BMA treatment. Moreover, the regression analysis demonstrates that decreasing or increasing the time length of BMA treatment prior to extraction does not influence the incidence of MRONJ.

The limitations of this study include a lack in available data about lifestyle habits, including smoking and alcohol consumption. Furthermore, the fact that a few different operators performed extractions might have led to some bias. We did not categorize the dental extractions by anatomical location. It was recently shown that MRONJ lesions in the maxilla are amenable to surgical treatment with a high success rate [34]. Anatomical location, however, was a confounder beyond the scope of this study. 

It seems that three major factors are involved in the development of MRONJ in cases of dental extraction while under BMA treatment: surgery, the least important and unavoidable risk factor, which involves changing tissue homeostasis and starting the wound healing cascade; local inflammation/infection, the most important risk factor, which involves avoiding the transition from the inflammatory to the proliferative stage; and BMA treatment, which amplifies the ability of local inflammation/infection to disturb the wound healing cascade. Therefore, despite the fact that starting BMA treatment post-extraction does not eliminate the possibility of MRONJ, it can still lower its incidence. 

Based on the findings of this retrospective cohort study, the following protocols for reducing the odds of the development of MRONJ in patients receiving high doses of BMAs are recommended: patients should be evaluated prior to BMA treatment for the necessity of tooth extraction. Local inflammation/infection should be minimized prior to surgery through local intervention, mouthwash, and/or local or systemic antibiotics. Extraction should be performed after the elimination of local inflammation/infection. A waiting time of at least 6 weeks is recommended to allow the soft tissue to finish the proliferative stage and enter the remodeling stage. BMA treatment can then be initiated, minimizing the chances of MRONJ. Such a protocol requires further validation in future prospective randomized studies. 

## 5. Conclusions

Local inflammation/infection and onset of BMA treatment prior to extraction were the only influencing variables, with 10.23 times the chance of developing MRONJ following tooth extraction. Future protocols should use this information to minimize MRONJ incidence. 

## Figures and Tables

**Table 1 jcm-11-01807-t001:** Demographic data for the 93 patients in the study group.

	Variable	No.	%
Sex	Men	21	29.17
	Women	72	70.83
Age (years)	Range	32–84	
	Average	62.16 ± 11.5	
Relevant medical diagnoses	Diabetes	13	13.97
	Anemia	2	2.93
Relevant medications	Anticoagulants	14	15.05
	Steroids	8	8.60
	Metformin	11	11.82
Cancer type	Breast	59	63.44
	Multiple myeloma	18	19.35
	Lung	3	3.23
	Prostate	3	3.23
	Other	10	10.75
BMA	Aredia	30	32.26
	Zomera	54	58.06
	Denosumab	9	9.68

**Table 2 jcm-11-01807-t002:** Dental extraction data.

		N	%
Dental extractions		103	100
Time frame between start of BMA and dental extraction (months)	>6 prior	30	29.13
	>2 and <6 prior	9	8.74
	<2 prior	14	13.59
	<2 after	5	4.85
	>2 and <6 after	10	9.71
	>6 after	35	33.98
Drug holiday (>2 months)		14	13.3
Number of teeth extracted	1	43	41.75
	2	23	22.33
	3	12	11.65
	4	8	7.77
	5	5	4.85
	≥6	12	11.65
Indication for extraction	Caries	34	33.01
	MRONJ	3	2.91
	Periapical pathosis	10	9.71
	Pericoronitis	2	1.94
	Periodontal disease	45	43.69
	VRF *	7	6.8
	Other	2	1.94
Local inflammation/infection		67	65.04
Post-extraction treatment	Antibiotics	2	1.94
	Mouthwash	23	22.33
	Both	78	75.73
Time of follow-up (years)	Range	1–7	
	Average	1.97 ± 1.38	
Development of MRONJ		20	19.42

* VRF: vertical root fracture.

**Table 3 jcm-11-01807-t003:** Relationship between categorical variables for the development of MRONJ in 103 extraction procedures.

Variables	Category	MRONJ		*p* Value
		No	Yes	
Total		83 (80.6%)	20 (19.4%)	
Sex	Men	19 (23%)	4 (20%)	0.522
	Women	64 (77%)	16 (80%)	
Cancer type	Breast	51 (61.4%)	14 (70%)	0.192
	Lung	3 (3.6%)	0	
	Multiple myeloma	16 (19.3%)	4 (20%)	
	Prostate	2 (2.4%)	2 (10%)	
	Other	11 (13.3%)	0	
Diabetes or anemia	Yes	70 (84.3%)	17 (85%)	0.62
	No	13 (15.7%)	3 (15%)	
Anticoagulants	Yes	14 (16.9%)	3 (15%)	>0.99
	No	69 (83.1%)	17 (85%)	
Steroids	Yes	7 (8.4%)	2 (10%)	>0.99
	No	76 (91.6%)	18 (90%)	
Metformin	Yes	9 (10.8%)	1 (5%)	0.682
	No	74 (89.2%)	19 (95%)	
BMA type	Zomera	50 (60.2%)	9 (45%)	0.308
	Aredia	27 (32.5%)	8 (40%)	
	Denosumab	6 (7.2%)	3 (15%)	
Drug holiday	Yes	6 (50%)	7 (50%)	0.652
	No	6 (50%)	7 (50%)	
Indication	Caries	32 (38.6%)	1 (5.3%)	0.01
	MRONJ	1 (1.2%)	2 (10.5%)	
	PA pathosis	6 (7.2%)	4 (21.1%)	
	Pericoronitis	2 (2.4%)	0	
	Perio	35 (42.2%)	10 (52.6%)	
	VRF	5 (6%)	2 (10.5%)	
	Other	2 (2.4%)	0	
Post-extraction treatment	Antibiotics	0	2 (%6.2)	0.115
	Mouthwash	16 (19.2%)	7 (25%)	
	Both	64 (80.8%)	14 (68.8%)	
Local inflammation	Yes	48 (58.5%)	19 (95%)	0.001
	No	34 (41.5%)	1 (5%)	
Onset of BMA treatment prior to extraction	Yes	36 (45.5%)	14 (74%)	0.025
	No	43 (54.5%)	5 (26%)	

**Table 4 jcm-11-01807-t004:** Relationship between numerical variables for the development of MRONJ in 103 extraction procedures.

		MRONJ		
Variable		NO	YES	
Age	Mean (SD)	61.73 (11.81)	64.1 (9.22)	0.406
Length of BMA treatment prior to extraction (Years)	Median (Q1, Q4)	2.25 (1, 5.75)	4.5 (1.87, 6)	0.432
Drug holiday (months)	Median (Q1, Q4)	4 (1.75, 36)	5 (2, 6)	>0.99
Number of teeth extracted	Median (Q1, Q4)	2 (1, 3)	1 (1, 3.25)	0.69

## Data Availability

Not applicable.

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
