# Peer review of "Minimizing MRONJ after Tooth Extraction in Cancer Patients Receiving Bone-Modifying Agents"

_jcm, 2022, doi:10.3390/jcm11071807_

Round 1

Reviewer 1 Report

Manuscript ID: jcm-1647899

Title: Minimizing MRONJ after tooth extraction in cancer patients receiving bone modifying agents

1.What is the main question addressed by the research?

To assess the ability of various suspected predisposing factors to predict medication-related osteonecrosis of the jaw (MRONJ) occurrence in patients on bone modifying agents (BMA) for oncology indications following tooth extraction.

2.Is it relevant and interesting?

The article is relevant and interesting.

3.How original is the topic?

The topic is current.

4.What does it add to the subject area compared with other published material?

The authors have collected and analyzed a great deal of recent data.

5.Is the paper well written?

Yes, the article is well written.

6.Is the text clear and easy to read?

Yes, but moderate English editing is required.

7.Are the conclusions consistent with the evidence and arguments presented?

Yes, the conclusions consistent with the evidence and arguments presented but further studies are necessary to confirm authors’ hypothesis.

8.Do they address the main question posed?

Yes, the Authors addressed the main question posed.

Other comments:

  • English language: Moderate English editing is required.
  • Abstract: To attract the reader's attention, please clarify the target of the article, and structure the abstract.
  • Introduction: This section needs some improvements. I recommend reading the Meeting Report of the first Workshop of European task force on MRONJ: article by Schiodt et al [PMID: 31325201]. Please refer to wide range of medications classified as tyrosine kinase inhibitors, monoclonal antibodies (including tocilizumab), mammalian target of rapamycin inhibitors, radiopharmaceuticals, selective estrogen receptor modulators, and immunosuppressants implicated in osteonecrosis of the jaws. I recommend Chang et al's article entitled "Current Understanding of the Pathophysiology of Osteonecrosis of the Jaw" to implement this section [PMID: 30155844]. Please refer to relationship between oncological symptoms (like numb chin syndrome) and MRONJ [PMID: 29680775]. Please refer to efficacy of drug holiday in MRONJ prevention [PMID: 32568416].
  • Materials and Methods: This section needs some improvements. Please follow CONSORT guidelines (http://www.consort-statement.org). First of there is no detailed description of surgeries. There are a lot of papers concerning how to perform dental extraction for MRONJ prevention. How were they performed?
  • Results: Please follow CONSORT guidelines (http://www.consort-statement.org). Please correct Tables.
  • Discussion: This section needs some improvements. Please discuss the use of platelet concentrates for MRONJ prevention with reference to a recent systematic review on the subject published in the EACMFS reference journal. Anyway, the use of platelet concentrate in oncological patient is at the center of a recent academic debate [PMID: 31116189]. Please discuss the efficacy of drug holiday in MRONJ prevention [PMID: 32568416].
  • Conclusion: This section was properly prepared but further studies are necessary to confirm authors’ hypothesis.

After making the indicated changes, I am available for a second round of peer review.

Thanks for the opportunity to review this manuscript.

Reviewer 2 Report

This study is good job and worth publishing in the journal if the minor points are safely corrected.

Abstract:
You wrote “Tooth 16 extraction is a significant risk factor for MRONJ” in abstract line 3, however, tooth extraction itself is no longer a strong risk factor. You have to arrange this mention here. I also suggested this in introduction section. Please check. 

Introduction:
L50: Now, simple tooth extraction is no longer a strong risk factor according to Soutome, et al. Using large cohort, they concluded tooth extraction during BMA therapy did not increase the risk. Their propensity score matching analysis showed that tooth extraction rather significantly lowered the risk of MRONJ development. They con concluded the problematic teeth that can be an infection source increases the risk of MRONJ, and thus, they need to be extracted even during BMA administration.
Soutome S, Otsuru M, Hayashida S, Murata M, Yanamoto S, Sawada S, Kojima Y, Funahara M, Iwai H, Umeda M, Saito T. Sci Rep. 2021;11(1):17226.
You can add this explanation in this section and should highlight that tooth extraction should not always be avoided in patients treated with BMAs. 

Word change recommendation:
L64: Group → Cohort
L75, 199: IV → 4

Table 1: Anticoagulants, Steroids, and Metformin are just a name of drug. So, they should not be included in “medical history” together. You can establish an additional item named “drugs”.
Moreover, I cannot believe 93 cancer patients in your cohort had only DM or anemia? They can have other medical history such as osteoporosis? If you did not include the osteoporosis patients, you must mention it in the exclusion criteria in Material and Methods section. 
Moreover, you must explain what cancers were included in “other”.   

Table 2: What is “PA pathosis”? This is not a general word/abbreviation, you have to explain what the words means.

Discussion:
L203: You can mention this result supports the result from Soutome, et al.(above mentioned). 

Your discussion is lack of any limitations of the study. In general, there is the difference on the development of MRONJ between maxilla and Mandible, and the number of cases of maxillary onset is much smaller than that of the mandible. Moreover, regarding the healing rates of MRONJ, Okuyama, et al. reported the high healing rate of maxillary MRONJ which were treated surgically (below). You should include your limitation that the tooth extraction site focusing on the anatomy (maxilla or Mandible, or anterior tooth or molars) is not in consideration in the study, citing the article below.
Okuyama K, Hayashida S, Rokutanda S, Kawakita A, Soutome S, Sawada S, Yanamoto S, Kojima Y, Umeda M.  Surgical strategy for medication-related osteonecrosis of the jaw (MRONJ) on maxilla: A multicenter retrospective study. J Dent Sci. 2021;16:885-890.
Limitations should also be included life style habits like smoking and alcoholic consumption.

In total, the English editing should be required before the publishment.

Round 2

Reviewer 1 Report

The Authors changes improved the manuscript quality.

On page 2, Line 76 Authors should insert a sentence regarding platelet concentrates that is really important on the topic:"Anyway, the use of platelet concentrate in oncological patient is controversial [PMID: 31116189]."

Author Response

Thank you for your swift response !

Your reference and citation have been added as per your recommendation.